# Target-dependent RNA polymerase as universal platform for gene expression control in response to intracellular molecules

Shodai Komatsu[1,2], Hirohisa Ohno[1] & Hirohide Saito [1,2] ✉

Controlling gene expression in response to specific molecules is an essential technique for regulating cellular functions. However, current platforms with transcription and translation regulators have a limited number of detectable molecules to induce gene expression. Here to address these issues, we present a Target-dependent RNA polymerase (TdRNAP) that can induce RNA transcription in response to the intracellular target specifically recognized by single antibody. By substituting the fused antibody, we demonstrate that TdRNAPs respond to a wide variety of molecules, including peptides, proteins, RNA, and small molecules, and produce desired transcripts in human cells. Furthermore, we show that multiple TdRNAPs can construct orthogonal and multilayer genetic circuits. Finally, we apply TdRNAP to achieve cell-specific genome editing that is autonomously triggered by detecting the target gene product. TdRNAP can expand the molecular variety for controlling gene expression and provide the genetic toolbox for bioengineering and future therapeutic applications.

Controlling gene expression in response to specific intracellular molecules is a powerful strategy for monitoring cellular conditions and regulating cellular programs[1,2]. In nature, metabolic systems use transcription and translation regulators to monitor the consumption of cellular metabolites and precisely regulate metabolic pathways[3]. Immune systems also use RNA and DNA sensors to detect foreign nucleic acids and subsequently activate immune-related genes[4,5]. These natural gene-regulatory components, such as Tet repressors and riboswitches, have been repurposed to control gene expression and cellular functions in an inducible manner[1,6,7]. However, natural transcription and translation regulators have limited molecules to induce gene expression. Moreover, most engineered systems are designed to respond to external stimulation by small molecules, such as doxycycline. Although directed evolution and computational design have altered the ligand specificity of these natural regulators, reengineering their ligand receptors with large conformational changes is still challenging[7,8]. Thus, there is a need for an autonomous gene-regulatory platform that can

respond to various intracellular biochemical information including proteins/peptides, and RNA, which reflect viral and disease pathogenesis, for subsequent cellular regulations and applications.

Antibodies are highly potent proteins capable of targeting a wide variety of molecules including proteins, RNA, and small molecules. While full-length antibodies have been used to target extracellular molecules, several small antibodies such as single-chain variable fragments (scFvs) and nanobodies have been optimized to target intracellular molecules[9–11]. These small antibodies are derived from the variable region of antibodies which consists of complementary determining regions (CDRs) and a framework region. The CDR sequence is variable, allowing the tunable molecular specificity of antibodies. The framework sequence is responsible for the intracellular stability of variable regions. Thus, antibody variable regions hold great potential as versatile ligand receptors in gene regulation. Previously, scFvs and nanobodies have been used to regulate gene transcription and translation in response to target proteins[12–14]. However,

[1]Department of Life Science Frontiers, Center for iPS Cell Research and Application, Kyoto University, 53 Kawahara-cho, Shogoin, Sakyo-ku, Kyoto 606-8507, Japan. [2]Graduate School of Medicine, Kyoto University, Yoshida-Konoe-cho, Sakyo-ku, Kyoto 606-8501, Japan. ✉e-mail: saitou.hirohide.8a@kyoto-u.ac.jp

these approaches require two scFvs or nanobodies with distinct binding sites against the same target to recruit transcription or translation regulators on target proteins. This requirement significantly limits the range of detectable molecules of antibodies. Moreover, these systems rely on cellular transcription and translation machinery to synthesize mRNA and proteins, thus limiting their application beyond the differences between eukaryotes and prokaryotes. Other drawbacks are that the transcription system restricts detectable molecules to those localized in the nucleus, and the translation system limits molecular outputs to proteins. To achieve broader applications based on molecular targeting by antibodies, it is desirable to develop a gene-regulatory system that relies on single antibody alone and is independent of cellular transcription machinery.

Bacteriophage T7 RNA polymerase (T7 RNAP) exhibits high transcriptional activity and is capable of synthesizing RNA from DNA templates without the need for additional factors[15]. The T7 RNAP can be divided into N- and C-terminal fragments that spontaneously assemble into functional RNAP[16,17]. Previously, the split T7 RNAP has undergone molecular evolution to suppress the spontaneous assembly, resulting in a proximity-dependent RNAP[18]. This evolved split RNAP has enabled inducible RNA transcription that is regulated by the protein-protein interaction (PPI) between fused proteins. The light and chemical-inducible dimerization domains allowed the split RNAP to be activated by blue light and small molecules, respectively[18,19]. Thus, this split RNAP holds great potential to control gene expression in response to intracellular molecules without relying on cellular machinery. However, the available PPI inducible molecules for regulating split RNAP activity have been limited to commonly used small molecules, such as rapamycin and abscisic acid, due to a lack of inducible dimerization proteins[18,19]. As a result, the split RNAP has been unable to respond to intracellular biochemical information, limiting the potential of split RNAP for gene regulation and its range of applications.

In this study, we present a target-dependent split T7 RNA polymerase (Target-dependent RNAP or TdRNAP) as a universal gene-regulatory platform that enables controlling gene expression and cellular functions in response to various intracellular molecules. TdRNAP uses only single antibody to induce the assembly of split T7 RNAP and can trigger RNA transcription in response to the molecular targets specifically recognized by the fused antibody. We showed the flexible target tunability of TdRNAP using many identified antibodies that recognize peptides, proteins, RNA, and small molecules, demonstrating key advantages in the design of molecular responsive gene regulation and highlighting its ability as a biochemical information converter (Fig. 1a). Moreover, we constructed mammalian orthogonal and multi-layer genetic circuits that can independently transduce multiple intracellular information into desired genetic outputs in the same cell. Finally, we show the potential application of TdRNAP for precise and cell-specific genome editing that is autonomously regulated by sensing the presence of target proteins with minimal off-target effects. The TdRNAP platform greatly simplifies the design of inducible gene expression systems and opens up new possibilities for precise control of gene expression and modulation of cellular pathways in biological research, bioengineering, and therapeutic applications.

## Results

### Design and characterization of TdRNAP

To control gene expression in response to various intracellular molecules, we reasoned that the variable region of antibodies could be used as a fusion protein of split RNAP. In this concept, we aimed to induce split RNAP assembly based on target binding by variable regions of single antibody, because it allows controlling gene expression using various identified antibodies without the screening of secondary antibodies for split RNAP assembly. To enable this strategy, we focused on the structural components of the variable region. The variable region consists of heavy and light chains and thus can be divided into two small

domains: heavy and light chain variable domains (VH and VL). Previously, in vitro studies have shown that some separately expressed VH and VL domains can form a stable heterotrimeric complex with their molecular targets[20]. This suggests that each VH and VL domain retains the binding affinity and specificity to molecular targets of their parental antibodies, even when they are not connected with disulfide bonds or flexible linkers. Therefore, we hypothesized that VH and VL domains could induce the assembly of split RNAP into a functional RNAP by binding to their molecular targets (Fig. 1a). We named this RNAP architecture "Target-dependent RNAP (TdRNAP)".

To test our hypothesis, we first designed a TdRNAP using the VH and VL domains of an anti-GCN4 antibody. The anti-GCN4 antibody recognizes a leucine zipper peptide of the yeast transcription factor GCN4 with nanomolar affinity, and its variable region has an optimized framework that allows protein folding without relying on intramolecular disulfide bonds, which ensures the structural stability in the intracellular reducing environment[21,22]. We fused the VH and VL domains to C-terminal and N-terminal fragments of split T7 RNAP, respectively, resulting in GCN4-dependent RNAP (GCN4-dRNAP) (Fig. 1b). To test the GCN4-dRNAP, we used enhanced green fluorescent protein (EGFP) fused with GCN4 peptide (EGFP-GCN4) as a molecular target and investigated whether the induction with EGFP-GCN4 induces split RNAP assembly. The constructs for GCN4-dRNAP were cotransfected into the human embryonic kidney 293FT cells along with a reporter plasmid encoding near-infrared fluorescent protein 670 (iRFP670) under the control of T7 promoter, and an induction plasmid encoding EGFP-GCN4 or EGFP. We then analyzed iRFP670 expression with fluorescent microscopy. The coexpression with EGFP-GCN4 resulted in a strong enhancement in iRFP670 fluorescence compared with EGFP, indicating that GCN4 binding to VH and VL domains induces split RNAP assembly (Fig. 1c). This GCN4-dependent reporter induction was observed even when the fusion orientation of VH and VL domains was reversed. The additional reporter assay with luciferase as the reporter gene showed that the GCN4-dRNAP exhibits slightly higher activity when the VH and VL domains are fused to the C-terminal and N-terminal RNAPs, respectively (Supplementary Fig. 1a). We thus used this fusion pattern (T7N-VL and VH-T7C) for subsequent experiments. We also assessed the effect of linker length on the activity of GCN4-dRNAP. Luciferase assay revealed minimal linker-length dependency, consistent with a previous report in which FK506 binding protein (FKBP) and FKBP-rapamycin binding domain (FRB) are fused to RNAP fragments (Supplementary Fig. 1b)[18]. We noted that T7 RNAP undergoes large conformational changes during the transition from an initiation complex to an elongation complex[23]. Because larger or highly charged molecular targets may interfere with such conformational changes, we decided to use the longer linker for subsequent experiments to minimize such negative effects.

Next, we investigated whether the activity of GCN4-dRNAP increases in a dose-dependent manner. To examine the dose dependency, we varied the induction levels of EGFP-GCN4 and conducted luciferase assays. The induction with EGFP-GCN4 resulted in a dose-dependent increase in luminescence signals, whereas EGFP without GCN4 did not increase the reporter signals even at high induction levels (Fig. 1d). Finally, we investigated whether the binding affinity between the antibody and molecular target affects the activity of TdRNAP. To examine this affinity dependency, we used three VH variants of the anti-GCN4 antibody with amino acid mutations in the CDR which decreased the binding affinity for the GCN4 peptide[22]. The luciferase assay showed the reduction in binding affinity decreases the RNAP activity (Fig. 1e). In particular, GFA, a VH variant with more than 300-fold lower affinity than WT[22,24], substantially decreased the RNAP activity. Whereas, GLW, with 1.7-fold lower affinity than WT[22], exhibited comparable activity to WT, suggesting that the dissociation constant of 0.6 nM is strong enough to maximize the activity of TdRNAP in mammalian cells. These results show that the activity of TdRNAP

depends on the dose of intracellular targets and the binding affinity between the fused antibody and the corresponding target. Taken together, our results demonstrate that the interaction between VH/VL domains and their molecular target can lead to the assembly of split RNAP and induce target-dependent gene transcription.

## Expanding the repertoire of molecular targets by antibody substitution

Next, we investigated whether antibody substitutions allow TdRNAP to induce reporter expression in response to the corresponding targets. To test this, we first used the VH and VL domains of an anti-FLAG antibody to design FLAG-dependent RNAP (FLAG-dRNAP) (Fig. 2a, top). The anti-FLAG antibody, clone M2, recognizes a FLAG octapeptide (DYKDDDDK) with nanomolar affinity ($K_d = 6.5$ nM)[25,26]. However, the intracellular stability of its variable region remains unclear. Upon the induction with FLAG-tagged EGFP (EGFP-1xFLAG), we found that the FLAG-dRNAP could induce reporter expression, but the RNAP activity was very low despite its high binding affinity (Supplementary

Fig. 2a). The RNAP activity was not significantly improved even when using three tandem FLAG tags (3xFLAG) which enhance the binding affinity[27] (Supplementary Fig. 2a). From these results, we hypothesized that the structure of the original variable region might be unstable in the reducing environment of the cytoplasm due to its dependence on intramolecular disulfide bonds for proper folding.

Previously, the VH and VL domains of the anti-GCN4 antibody have been optimized for intracellular expression by grafting their CDRs onto framework regions of stable antibodies[10,11]. Based on this approach, we grafted the CDRs of the anti-FLAG antibody onto a stable framework region derived from an anti-HER2 antibody, trastuzumab[11] (Supplementary Fig. 2b). As a result, we found that the CDR grafting of the VL domain successfully enhanced the activity of FLAG-dRNAP, indicating that the structure of the VL domain of the anti-FLAG antibody was unstable within the cells (Fig. 2b). The CDR grafting of both VH and VL domains further increased reporter expression, but it also induced leak expression in the absence of the FLAG tag, suggesting that the grafted VH and VL domains of the anti-FLAG antibody weakly

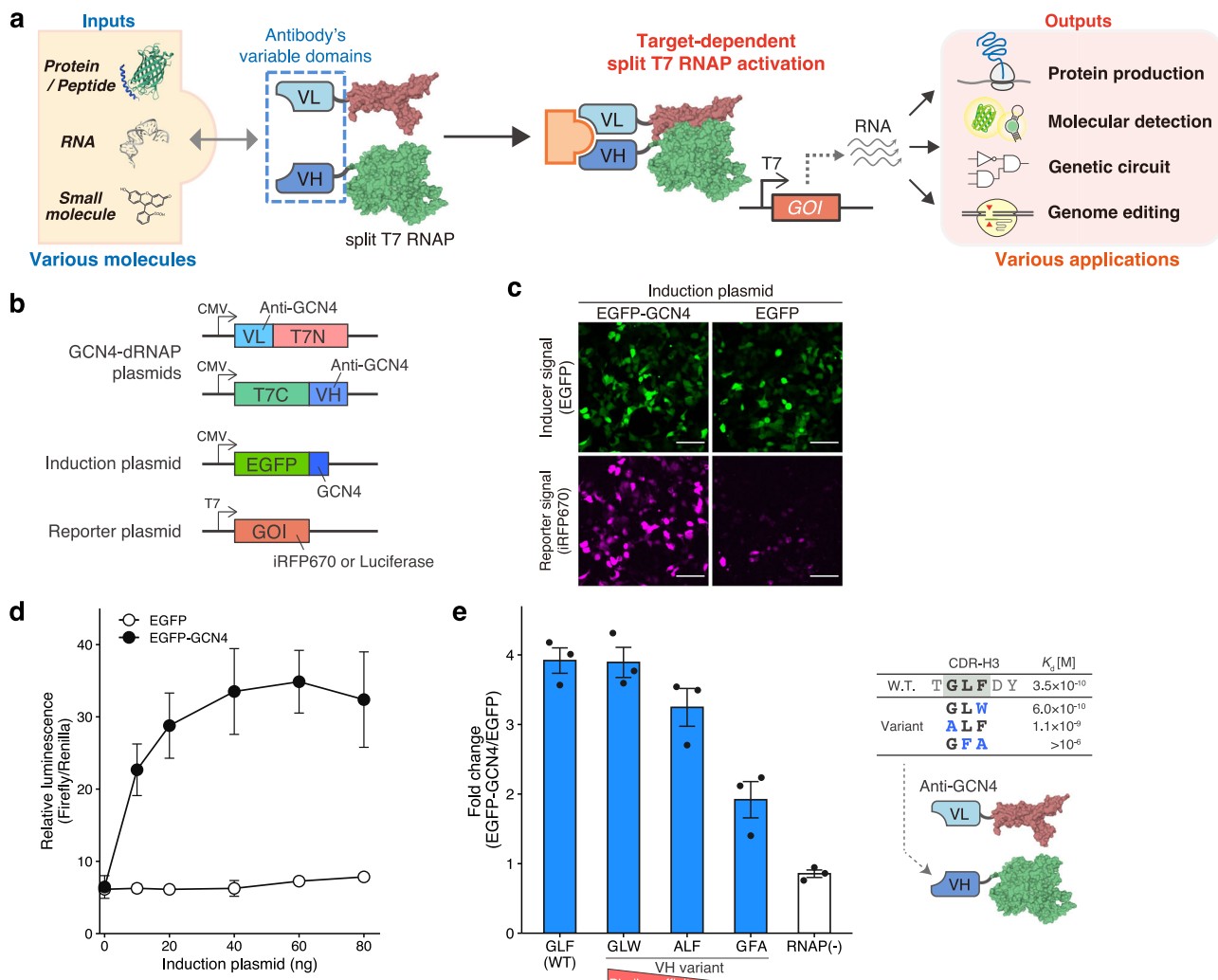

**Fig. 1 | Design and characterization of TdRNAP. a** Schematic of TdRNAP design and strategy. Split T7 RNAP assembles into a functional RNAP when the fused VH and VL domains interact with molecular targets. The activated RNAP can transcribe genes of interest (GOI) under the control of T7 promoter, resulting in various outputs and applications. **b** Plasmids used to test GCN4-dRNAP in 293FT cells. N-terminal and C-terminal split RNAP fragments (T7N and T7C) were fused to VL and VH domains of the anti-GCN4 antibody, respectively. **c** Fluorescence images of 293FT cells transfected with the GCN4-dRNAP and induced with EGFP or EGFP-GCN4. Scale bar, 100 μm. **d** Dose-dependent transcriptional activation of GCN4-dRNAP. 293FT cells were transfected with the reporter plasmid encoding firefly luciferase and *Renilla* luciferase under the control of T7 and constitutive promoters, respectively. Cells were then analyzed for luminescence. **e** Dependency of the antibody binding affinity on the transcription activity of GCN4-dRNAP (left). CDR-H3 sequence and the reported dissociation constant ($K_d$) of each VH variant (right). The protein and RNA structures were drawn with PDB data (PDB ID: 1QLN, 2Y0G, 1HLL, and 4PHY) (**a, e**). Fold changes are calculated from the induction with EGFP-GCN4 versus EGFP for each VH variant. Values represent mean ± s.e. of $n = 3$ biological replicates (**d, e**). Source data are provided as a Source Data file.

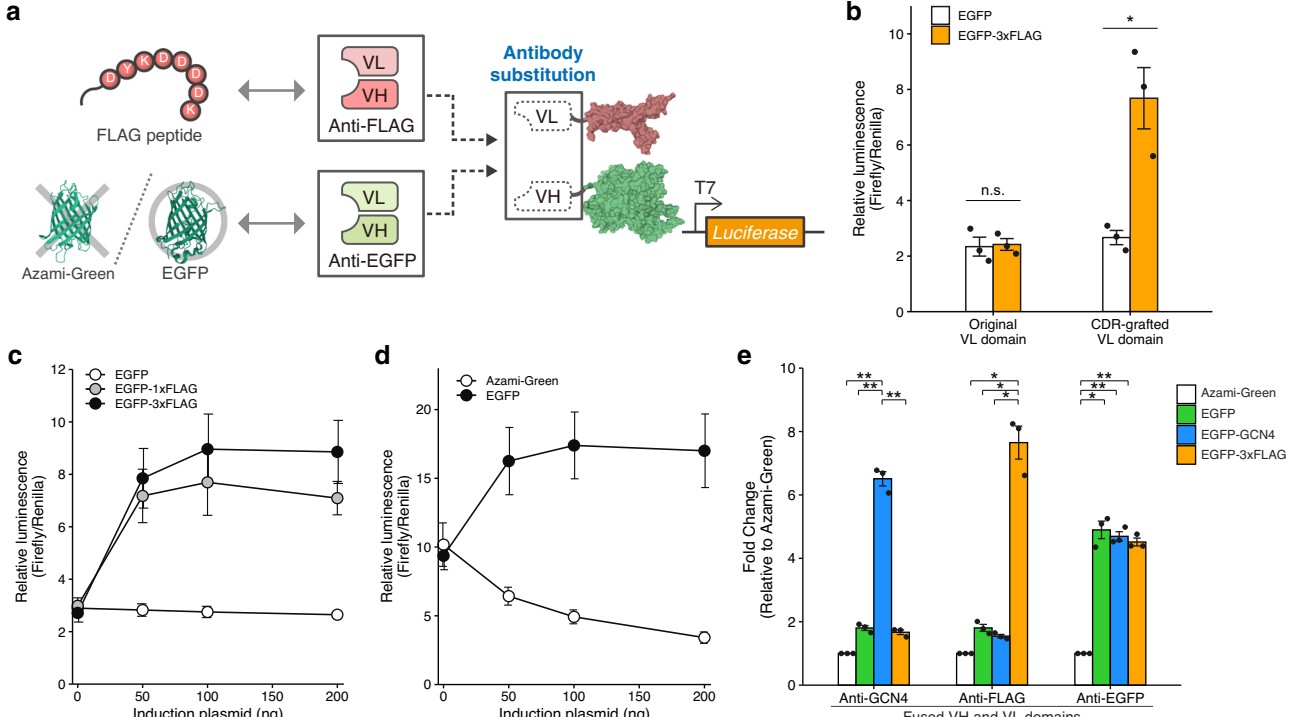

**Fig. 2 | Expanding the molecular repertoire for TdRNAP by antibody substitution.** **a** Antibody substitution of TdRNAP using the anti-FLAG and anti-EGFP antibodies, resulting in FLAG- and EGFP-dRNAPs. **b** Optimization of the framework region of VL domain by CDR grafting. CDRs for the VL domain of the anti-FLAG antibody were grafted to the trastuzumab framework region (CDR-grafted VL domain). **c** Dose- and affinity-dependent transcriptional activation of FLAG-dRNAP. **d** Target-specific and dose-dependent transcriptional activation of EGFP-dRNAP. **e** Validation of the target specificity of GCN4-, FLAG-, and EGFP-dRNAPs by inducing with their corresponding molecular targets. The protein structures were drawn with PDB data (PDB ID: 1QLN, 2Y0G, and 3ADF) (**a**). Values represent mean ± s.e. of *n* = 3 biological replicates (**b**–**e**). Statistical analysis by unpaired two-tailed *t* test (**b**) and one-way ANOVA with Bonferroni correction (**e**), *$P$ <0.05, **$P$ < 0.01, n.s., not significant ($P$ >0.05). Each $P$ value is listed in Supplementary Table 5. Source data are provided as a Source Data file.

associated with each other (Supplementary Fig. 2c). We thus used the FLAG-dRNAP with the grafted VL domain for subsequent experiments. We further investigated whether the optimized FLAG-dRNAP could exhibit dose- and affinity-dependent increases in the RNAP activity. Upon the induction with EGFP-1xFLAG, we observed a dose-dependent increase in reporter signals (Fig. 2c). The reporter signals were further enhanced upon the induction with EGFP-3xFLAG, indicating that the RNAP activity is affinity-dependent. These observations are consistent with the results obtained with GCN4-dRNAP (Fig. 1d, e). These results demonstrated that antibody substitution allows TdRNAPs to alter their molecular targets. Additionally, even if the variable regions of antibodies were unstable within the cells, we showed that the variable regions can be stabilized and adapted for TdRNAP using stable framework regions.

Based on the insight from FLAG-dRNAP, we designed an additional protein-dependent RNAP targeting larger proteins rather than small peptides. We selected an EGFP antibody for the next substitution because the anti-EGFP antibody had been designed based on the trastuzumab stable framework[28] (Fig. 2a, bottom). To examine the target specificity of EGFP-dependent RNAP (EGFP-dRNAP), we used EGFP and monomeric Azami-Green. Azami-Green is a green fluorescent protein that has a similar beta-barrel structure to EGFP but a low sequence identity of 29%. The induction with EGFP successfully increased the reporter signals in a dose-dependent manner, whereas the induction with Azami-Green did not increase the reporter signals even at high induction levels (Fig. 2d). These results clearly show that EGFP-dRNAP activates through the specific recognition of EGFP.

In addition, we tested whether each TdRNAP could specifically respond to their corresponding targets without a cross-reactivity. To examine the cross-reactivity, we used GCN4-, FLAG-, and EGFP-

dRNAP, and coexpressed them with individual targets. The luciferase assay showed that each TdRNAP was activated only in response to the corresponding target (Fig. 2e), demonstrating that the fused antibody variable regions specifically recognize their corresponding targets in mammalian cells. These findings show that each VH and VL domain retains its target specificity in mammalian cells, indicating that the CDR loop structures of the antibody variable regions are properly folded and stabilized. This suggests that the stable framework region and the molecular chaperone of mammalian cells may contribute to the stabilized CDR loop structures and target specificity. Taken together, these results demonstrate that antibody substitution enables the expansion of intracellular targets for TdRNAP, and the substituted antibodies retain the high specificity to the corresponding targets.

**Design of RNA and small-molecule-dependent RNAPs**
Our results raised the question of whether TdRNAP could regulate reporter expression in response to other types of molecules, such as RNA and small molecules. Several RNA sensing technologies have already been developed based on engineered RNAs[29–31]. However, these technologies rely on complementary base pairing for RNA targeting, making it challenging to access structured RNAs where target sequences are masked by RNA secondary and tertiary structures. In contrast, antibodies exhibit structure specificity for RNA recognition rather than sequence specificity[32,33]. This structure specificity could be advantageous when targeting viral RNA because viruses have high mutation rates but maintain conserved structured regions within their genome, which is essential for viral replication and packaging the genomic RNA. We thus aimed to design an RNA-dependent RNAP (RNA-dRNAP) using an anti-viral RNA antibody and sought to

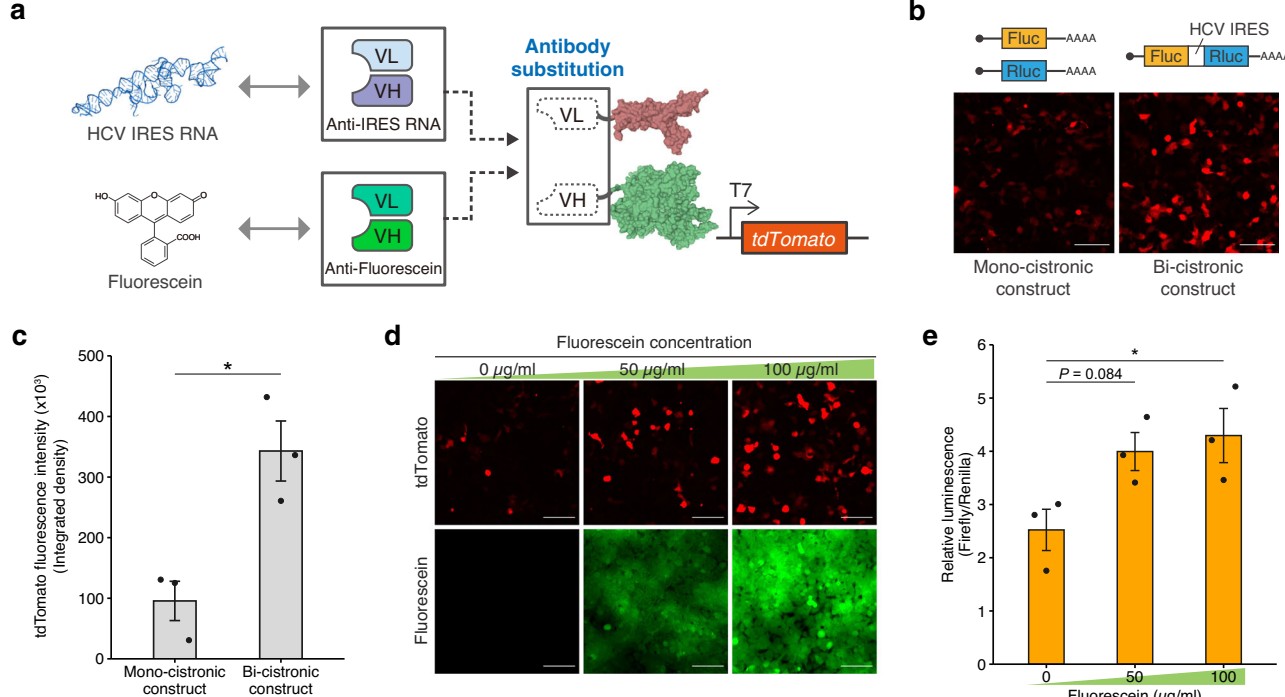

**Fig. 3 | Design and characterization of RNA and small-molecule-dependent RNAPs. a** Design of RNA and small-molecule-dependent RNAPs using the anti-HCV IRES RNA and anti-fluorescein antibodies. **b** Fluorescence images of 293FT cells transfected with HCV IRES RNA-dRNAP and induced with mono-cistronic or HCV IRES-containing bi-cistronic constructs. Scale bar, 100 μm. **c** Comparison of the tdTomato fluorescence intensity between cells induced with mono-cistronic and bi-cistronic constructs. The fluorescence intensity is quantified from the integrated density of tdTomato fluorescence. **d** Fluorescence images of 293FT cells transfected with Fluorescein-dRNAP and induced with 0, 50, or 100 μg/ ml fluorescein. Scale bar, 100 μm. **e** Dose-dependent transcriptional activation of Fluorescein-dRNAP. 293FT cells were transfected with the reporter plasmid encoding firefly luciferase and *Renilla* luciferase under the control of T7 and constitutive promoters, respectively. The protein and RNA structures were drawn with PDB data (PDB ID: 1QLN and 5A2Q) (**a**). Values represent mean ± s.e. of *n* = 3 biological replicates (**c**, **e**). Statistical analysis by unpaired two-tailed *t* test (**c**) and one-way ANOVA with Dunnett's multiple comparison test against the 0 μg/mL Fluorescein (**e**), *$P$ < 0.05. Each $P$ value is listed in Supplementary Table 5. Source data are provided as a Source Data file.

investigate whether RNA-dRNAP can detect structured regions of viral RNA within mammalian cells.

To test this, we used an anti-HCV IRES RNA antibody that specifically recognizes a structured RNA region within the internal ribosome entry site (IRES) of the hepatitis C virus (HCV) genomic RNA[34]. The VH and VL domains of the anti-HCV IRES RNA antibody were fused to split RNAP fragments, resulting in IRES RNA-dRNAP (Fig. 3a, top). The constructs for IRES RNA-dRNAP were cotransfected along with a reporter plasmid and a bi-cistronic construct with HCV IRES encoding firefly and Renilla luciferases. We then conducted fluorescent reporter assays using tdTomato as the reporter gene. Interestingly, we observed the robust enhancement of tdTomato fluorescence upon the induction with IRES-containing bi-cistronic mRNA (Fig. 3b, c and Supplementary Fig. 3a) whereas the induction with mono-cistronic mRNAs without HCV IRES exhibited lower tdTomato fluorescence. We simultaneously conducted luciferase assays and confirmed the Renilla luciferase expression from HCV IRES of the bi-cistronic construct, which demonstrates that HCV IRES properly folds into the functional structure in the living cells. The luciferase assays also showed that the expression levels of firefly and Renilla luciferases were comparable between the bi-cistronic and mono-cistronic constructs, indicating that translated luciferases did not affect the RNAP activity (Supplementary Fig. 3b). These results demonstrate that the IRES RNA-dRNAP activates transcription through the detection of the IRES region of HCV in living cells.

We next investigated whether TdRNAP could respond to small molecules in living cells using anti-small-molecule antibodies. To test this, we employed an anti-fluorescein antibody and designed Fluorescein-dependent RNAP (Fluorescein-dRNAP) (Fig. 3a, bottom)[35].

The constructs for Fluorescein-dRNAP were transfected into cells along with the reporter plasmid encoding tdTomato. We then treated the transfected cells with fluorescein diacetate, which is converted into fluorescein within living cells, and performed fluorescent reporter assays. Treatment with fluorescein led to a dose-dependent increase in reporter expression, while the treatment with the vehicle only mildly induced reporter expression (Fig. 3d, e). These results show that fluorescein promotes the assembly of the VH and VL domains and the activation of split RNAP. The results also indicate that the VH and VL domains exhibited a weak association even in the absence of fluorescein. The VH and VL domains of the anti-fluorescein antibody contain several residues in their CDRs that interact with each other to form a binding pocket for fluorescein loading[35]. This interaction might induce the spontaneous assembly of split RNAP even without fluorescein treatment.

In summary, anti-RNA and anti-small-molecule antibodies can also be adapted for use with TdRNAP. Thus, TdRNAP can serve as a versatile platform for controlling gene expression in response to a broad spectrum of biochemical molecules within mammalian cells.

### Constructing multilayer genetic circuits with TdRNAPs

Our results showed that TdRNAPs can serve as biochemical information converters that transduce various inputs to desired genetic outputs. This could provide versatile tools for constructing synthetic genetic circuits in living cells. We next sought to apply TdRNAPs to develop multilayer genetic circuits that can simultaneously control multiple gene expressions depending on the corresponding targets. We first designed a transcriptional amplification system using a T7 RNAP variant, CGG-R12-KIRV (CGG RNAP), as an

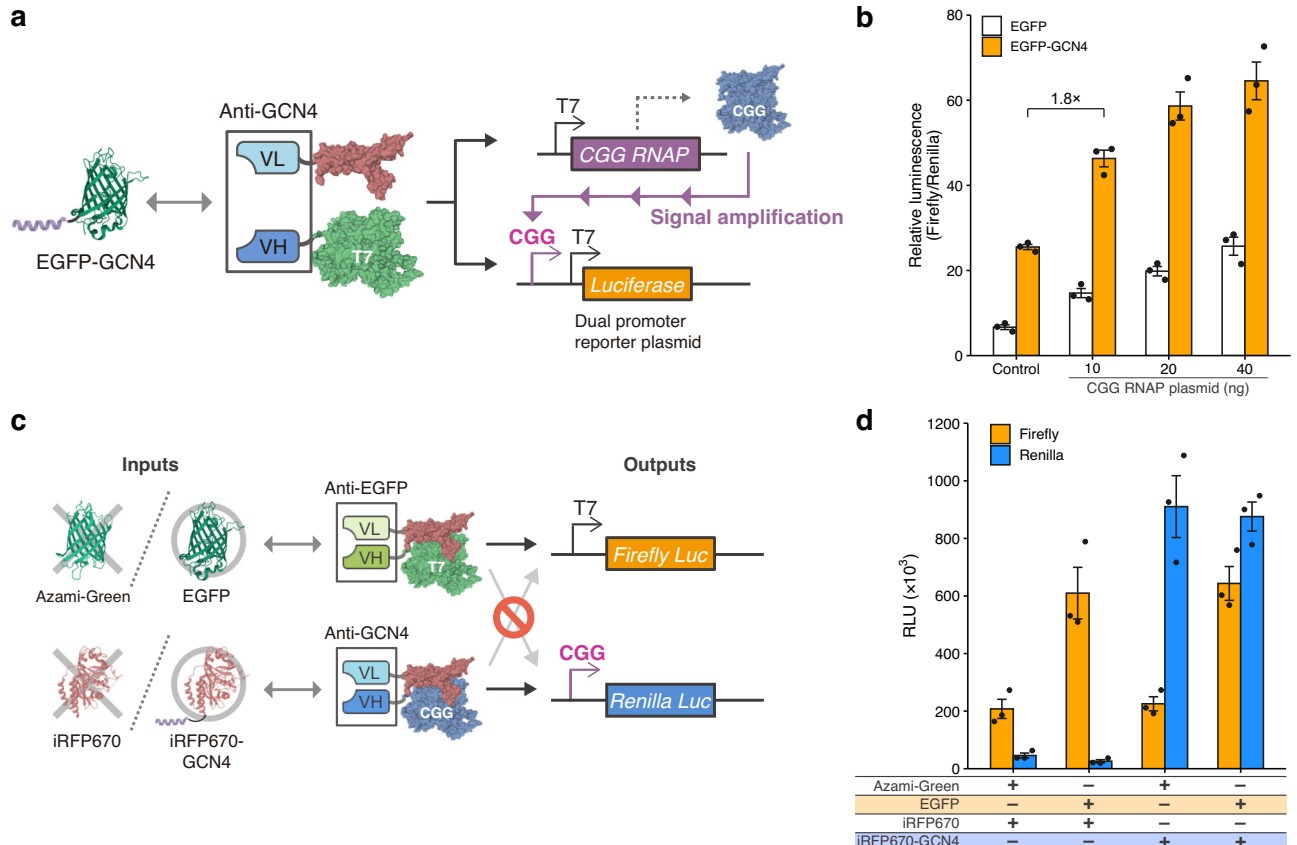

**Fig. 4 | Construction of multilayer genetic circuits with TdRNAPs for signal amplification and orthogonal signal transduction. a** Design of a transcriptional amplification system using CGG RNAP as an additional output of GCN4-dRNAP. The system involves a T7 promoter-driven CGG RNAP plasmid and a dual promoter reporter plasmid encoding a luciferase reporter gene under the control of both T7 and CGG promoters. The activated GCN4-dRNAP induces the expression of both CGG RNAP and the reporter gene. The reporter gene expression is then further enhanced by the induced CGG RNAP. **b** Validation of the enhancement of reporter expression by the amplification system in 293FT cells. **c** Design of an orthogonal genetic circuit consisting of EGFP-dependent T7 RNAP and GCN4-dependent CGG RNAP. Each TdRNAP independently controls the expression of a luciferase reporter gene under its corresponding promoter. EGFP-dependent T7 RNAP and GCN4-dependent CGG RNAP correspond to the expression of firefly and *Renilla* luciferases, respectively. **d** Simultaneous monitoring of target-dependent reporter expressions in the same 293FT cells. RLU, relative light units. The protein structures were drawn with PDB data (PDB ID: 1QLN, 2Y0G, 1P4B, 3ADF, and 5VIV) (**a**, **c**). Values represent mean ± s.e. of *n* = 3 biological replicates (**b**, **d**). Source data are provided as a Source Data file.

additional layer of a reporter gene. CGG RNAP is an evolved T7 RNAP with amino acid mutations in the C-terminal region that allow RNAP to specifically recognize a mutated T7 promoter, called CGG promoter[36]. We reasoned that CGG RNAP under the control of T7 promoter could be used to amplify reporter gene expression by adding CGG promoter to our T7 promoter-driven reporter plasmids, which is advantageous for sensing the low amount of intracellular molecules (Fig. 4a). We thus inserted CGG promoter upstream of T7 promoter in our reporter plasmid, resulting in a dual promoter reporter plasmid encoding luciferase under the control of T7 and CGG promoters.

To test this amplification system, we used GCN4-dRNAP and examined whether the reporter signals were enhanced even at low induction levels of EGFP-GCN4. The constructs for GCN4-dRNAP were transfected into 293FT cells along with the dual promoter reporter plasmid and the additional reporter plasmid encoding CGG RNAP. Upon the induction with 10 ng of the EGFP-GCN4 expression plasmid, the amplification system exhibited a 1.8-fold enhancement of reporter expression compared with the nonamplification system (Fig. 4b). Notably, the amplified luminescence signal was 1.3-fold higher than the saturated value of the nonamplification system which was observed at the sixfold higher induction level of EGFP-GCN4 (60 ng) (Figs. 1d and 4b). The amplification system further enhanced the reporter signals depending on the amount of T7 promoter-driven

CGG RNAP plasmid, although the background levels were also increased. These observations show that the additional reporter layer of CGG RNAP plays the role of a signal amplifier and is able to amplify the reporter signals even at low concentrations of the molecular target.

Next, we designed an orthogonal genetic circuit using two TdRNAPs that can simultaneously respond to two different targets as inputs and independently control two reporter genes as outputs. As a proof of concept, we designed the orthogonal system consisting of EGFP- and GCN4-dRNAPs. To independently control two reporter genes, EGFP- and GCN4-dRNAPs were constructed with split T7 and CGG RNAP, respectively, resulting in EGFP-dependent T7 RNAP and GCN4-dependent CGG RNAP (Fig. 4c). We investigated whether these two TdRNAPs could exhibit orthogonal gene regulation in a target-dependent manner using firefly and Renilla luciferase genes under the control of T7 and CGG promoters, respectively. The luciferase assay showed that the orthogonal system precisely controls the expressions of two reporter genes depending on the induction patterns of corresponding targets (Fig. 4d). This result demonstrates that two TdRNAPs simultaneously control their respective reporter genes in parallel. Taken together, these results show that the combination of antibodies and T7 RNAP variants expands the pattern of output signals and provides versatile tools for constructing multilayer genetic circuits in living cells.

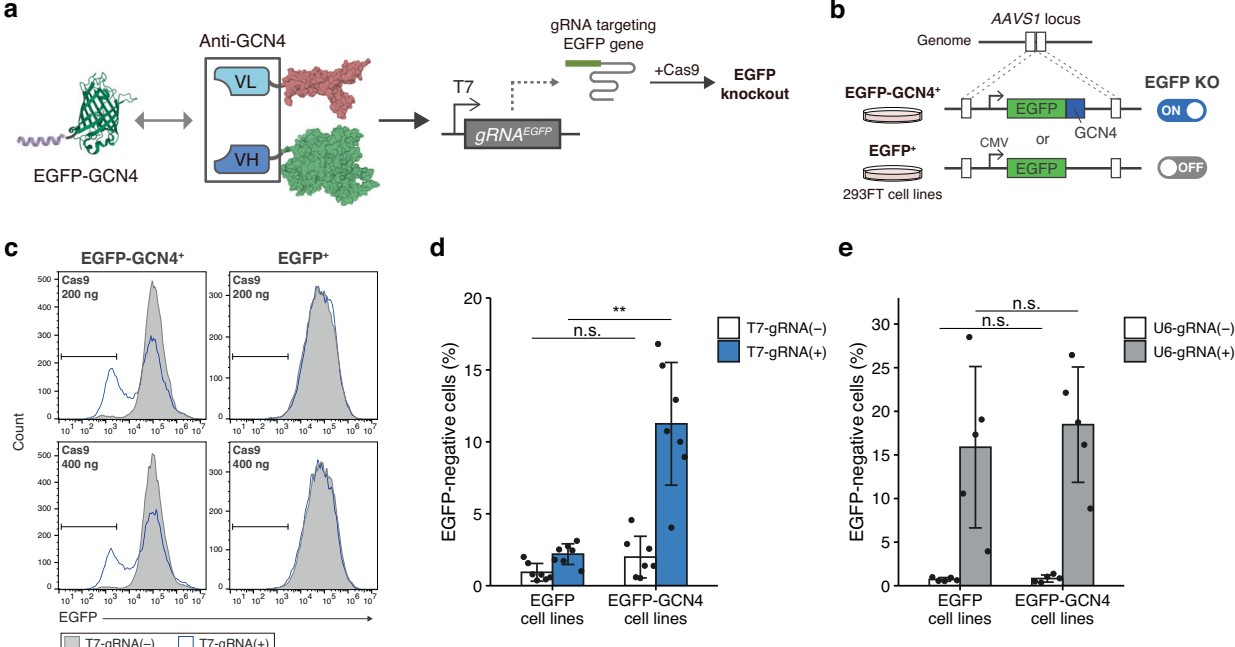

**Fig. 5 | TdRNAP enables cell-specific genome editing triggered by the detection of the target gene product. a** Application of TdRNAP to control genome editing in a target-dependent manner in human cells. GCN4-dRNAP induces gRNA expression when cells express GCN4-fused EGFP (EGFP-GCN4). The gRNA targets and knocks out the EGFP gene by forming a complex with Cas9. **b** Establishment of the 293FT cell lines carrying EGFP or EGFP-GCN4 gene in the *AAVS1* locus. GCN4-dependent EGFP knockout should be triggered preferentially in the cells expressing EGFP-GCN4 than EGFP. **c** Flow cytometry analysis of the GCN4-dependent EGFP knockout in the established 293FT cell lines. **d, e** EGFP-negative cell population measured by flow cytometry in EGFP knockout experiments using T7 promoter for GCN4-dependent gRNA expression (**d**) and U6 promoter for constitutive gRNA expression (**e**) along with 200 ng of Cas9 expression plasmid. The protein structures were drawn with PDB data (PDB ID: 1QLN, 2Y0G, and 1P4B) (**a**). Values represent mean ± s.d. of $n = 7$ (**d**) and $n = 5$ (**e**) biological replicates. Statistical analysis by unpaired two-tailed *t* test (**d, e**), **$P < 0.01$, n.s., not significant ($P > 0.05$). Each *P* value is listed in Supplementary Table 5. Source data are provided as a Source Data file.

## Target-dependent genome editing in human cells

The control of genome editing between targeted and non-targeted cells is one of the major challenges in gene therapy. To achieve such cell-specific genome editing, several biomarkers, including cell surface proteins and microRNAs, have been utilized to regulate the delivery and expression of genome editing machinery[37–39]. However, many of these biomarkers are by-products with elevated expression levels during disease progression, and the available biomarkers for the conventional approaches are still limited. Ideally, genome editing should be autonomously driven by detecting gene products derived from target genes with genetic alterations. In this strategy, TdRNAP has the advantage of intracellular target selectivity to control gene expression. Moreover, TdRNAP can generate functional RNA, such as guide RNA (gRNA), as outputs for genome editing. We thus hypothesized that TdRNAP could autonomously trigger genome editing by directly recognizing disease-associated proteins, such as mutant and fusion proteins, expressed from target genes in the genome.

To prove this concept, we designed a gene knockout experiment with CRISPR-Cas9 systems in which DNA cleavage was controlled by TdRNAP and preferentially induced in cells expressing a fusion gene (Fig. 5a). We aimed to induce EGFP knockout in a GCN4-dependent manner using GCN4-dRNAP and an EGFP-targeting gRNA under the control of T7 promoter. To test this, we used EGFP-GCN4 and EGFP genes as fusion and intact genes, respectively, and established two 293FT cell lines carrying EGFP-GCN4 or EGFP gene in the *AAVS1* locus (Fig. 5b). The constructs for GCN4-dRNAP and gRNA were cotransfected along with a Cas9 expression plasmid into EGFP-GCN4 and EGFP cell lines. We then measured the EGFP-negative cell population by flow cytometry. Flow cytometry analysis showed that GCN4-dRNAP induced EGFP knockout more preferentially in EGFP-GCN4 cell lines than EGFP cell lines, demonstrating that EGFP-GCN4 itself was used as a driver to knock out its own gene (Fig. 5c, d). Notably, we did not

observe the increase in EGFP-negative cell population in EGFP cell lines even at high Cas9 expression levels (Fig. 5c). These results indicate that the EGFP knockout event is tightly regulated by GCN4-dRNAP through controlling gRNA expression depending on intracellular EGFP-GCN4. For comparison, we used a constitutive gRNA expression plasmid with U6 promoter and conducted the same knockout experiment in which gRNA expression is independent of the GCN4 fusion. The constitutive gRNA expression resulted in EGFP knockout with comparable efficiency between two cell lines, indicating that the insertion of the nucleotide sequence for GCN4 peptide does not affect the knockout efficiency (Fig. 5e). These results demonstrate that TdRNAP can autonomously trigger gRNA expression by direct recognition of aberrant proteins expressed from target genes in the genome, enabling precise genome editing in a cell-specific manner.

## Discussion

Inducible control of gene expression is widely used for investigating biomolecular mechanisms and cell signaling pathways, constructing synthetic genetic circuits, and regulating cellular functions. Despite the broad range of applications, the variety of intracellular molecules that can induce gene expression has still been limited. In this study, we present TdRNAP as an universal platform for controlling gene expression in response to a wide variety of intracellular molecules. We showed that variable domains of single antibody can induce split T7 RNA polymerase assembly by binding to their corresponding targets. Using various identified antibodies against proteins/peptides, RNA, and small molecules, we demonstrated that TdRNAP can expand molecular repertoire to induce gene expression. To our knowledge, this is the first demonstration of transducing these three different types of biochemical information into transcriptional and translational outputs using a single platform. In particular, our IRES RNA-dRNAP presents the first example of transducing RNA structural information into RNA

synthesis. We also applied TdRNAPs to construct multilayer genetic circuits for signal amplification and orthogonal signal transduction. Finally, we demonstrated cell-specific genome editing that TdRNAP autonomously triggers gene knockout by detecting intracellular gene products derived from the target gene in the human genome.

The inducible gene expression has progressed based on the modification of natural transcription regulators. The transcription regulators with new molecular specificity have been engineered through directed evolution and computational design[40,41]. However, both approaches typically require the protein scaffolds for remodeling their molecular recognition motifs. As a result, the target tunability is significantly limited by the molecular preference of parental recognition motifs. In translational regulation, several RNA-based switches have been engineered through the randomization of natural riboswitches or in vitro aptamer screening[42,43]. However, the development of RNA-based switches with in vivo functional RNA-ligand pairs is still challenging. In contrast, our results demonstrated that TdRNAP has the advantage of using many identified antibodies to expand molecular targets for gene regulation. Because the methods for antibody screening are well-established, the antibodies with new molecular specificity can be easily identified and applied to TdRNAP. Furthermore, machine learning has facilitated antibody design[44]. This artificial intelligence (AI)-based method can further drive the development of TdRNAP.

Previously, split protein and inducible dimerization systems have been used to control gene expression[45–47]. In contrast to these approaches, TdRNAP has three advantages in design principle and regulation mechanism. First, it is possible to use VH and VL domains as inducible dimerization domains without selecting the split sites of ligand-binding proteins and screening dimerization domains. This design principle allows for the easy targeting of various intracellular molecules compared with conventional platforms, enabling gene regulation based on cellular conditions. This advantage greatly expands the application range for molecular-inducible gene regulation systems. Second, multiple and orthogonal TdRNAPs can be easily designed using reported T7 RNAP variants. The five orthogonal T7 RNAP variants and their respective promoters have been developed by directed evolution and are also available for split RNAP architecture[48,49]. These orthogonal RNAP and promoter pairs can be combined with different antibodies to design orthogonal TdRNAPs. Additionally, other single-subunit RNAPs derived from bacteriophages, such as SP6 RNAP, could also be used to design new orthogonal TdRNAPs by molecular evolution[50]. TdRNAPs designed with these evolved and orthogonal RNAPs could allow the multiplex molecular response and orthogonal gene regulation. Third, TdRNAP can be applied to both in vivo and in vitro studies. Because T7 RNAP itself has transcriptional activity, TdRNAP can be used in both bacteria and mammalian cells without the reliance on cellular transcription factors. This feature is also advantageous for in vitro studies like biosensor development and artificial cell creation[51,52]. One major limitation of TdRNAP is the requirement of high-affinity antibodies for high transcriptional activity. Our results indicate that antibodies with $K_d$ in the nanomolar to sub-picomolar range are preferable for designing TdRNAP with high performance in mammalian cells (Fig. 1e). To achieve a high dynamic range, similar to well-characterized inducible dimerization systems such as rapamycin-inducible systems[18,19,53], further improvement and optimization of the antibody-based dimerization system are likely necessary. However, this limitation could be improved by the affinity maturation combined with TdRNAP and AI-based antibody optimizations.

The key advantage of TdRNAP is not only the input tunability but also the output flexibility through the regulation of transcription and subsequent translation. We showed that TdRNAP produces both coding and noncoding RNAs for protein production and genome editing (Fig. 5). Moreover, we demonstrated that the combination of antibodies and T7 RNAP variants diversifies the output signal patterns by showing signal amplification and multiple signal transduction (Fig. 4). Overall, TdRNAP has great potential as a biochemical information converter for constructing synthetic genetic circuits and manipulating cellular gene networks.

TdRNAP could provide a potent toolbox for therapeutic applications. We presented the first demonstration of autonomous genome editing where the target gene product is used as a driver to knock out its own gene (Fig. 5). This result suggests that TdRNAP can provide new approaches in gene therapy to correct genetic mutations in a cell-specific manner. In particular, TdRNAP could provide a promising approach for the treatment of genetic diseases with limited sites for gRNA targeting, as well as those with higher and potential off-target risks[54,55]. The strategy targeting aberrant translational products could also be applied to eliminate cancer and aging cells because stop codon mutations and translational readthrough beyond stop codons are increased in aging and cancer, resulting in additional C-terminal tails derived from noncoding sequences[56,57]. Our target-dependent genome editing results also provide insight into the intracellular performance of TdRNAP for therapeutic applications. We showed that TdRNAP induces genome editing in response to the gene product expressed from the single genomic locus, which demonstrates that TdRNAP possesses sufficient sensitivity to intracellular gene products with more moderate and homogenous expression levels rather than transient overexpression through plasmid transfection. Furthermore, our initial research revealed that TdRNAP designed with an anti-Heat shock protein 70 (Hsp70) antibody[58,59] responds to the increase in endogenous Hsp70 expression after the heat shock stimulation (Supplementary Fig. 4), suggesting that TdRNAP could regulate the transcriptional activity depending on the expression levels of endogenous proteins. The dose-dependent activity may prove useful in targeting cancer cells with abnormal gene expression, without affecting healthy cells[60]. The ability to control gene expression may open up new possibilities for personalized medicine and precision therapies. We anticipate that TdRNAP provides a robust and versatile strategy for engineering genetic circuits and cellular function in both bioengineering and therapeutic applications.

## Methods
### Plasmid construction
The plasmids for each TdRNAP were constructed by HiFi assembly (NEB, #E2621) with PCR products or a Kunkel mutagenesis method with a pre-constructed plasmid. The genes for N-terminal and C-terminal fragments of T7 RNAP were amplified from plasmid pCAG-T7pol (Addgene plasmid #59926). The gene for the evolved T7 RNAP N-terminal fragment, d5-19, was generated by multiple site-directed mutagenesis. The genes for VH and VL domains of anti-GCN4 antibody were amplified from plasmid pHR-scFv-GCN4-sfGFP-GB1-NLS-dWPRE (Addgene plasmid #60906). The genes for anti-EGFP and anti-FLAG antibodies were amplified from synthetic DNA fragments (Thermo Fisher Scientific). The genes for anti-HCV IRES RNA, anti-fluorescein, and anti-Hsp70 antibodies were generated by the Kunkel mutagenesis method using the anti-FLAG antibody gene as the template DNA[61,62]. All plasmids were constructed by HiFi assembly with PCR products and are listed in Supplementary Table 1. All the standard and mutagenic PCR were conducted using PrimeSTAR Max DNA Polymerase (Takara Bio, #R045A). Amino acid sequences of individual proteins and peptides are listed in Supplementary Table 2. Nucleotide sequences of key genes and regulatory elements are listed in Supplementary Table 3.

### Cell culture and stimulation
293FT cells (Thermo Fisher Scientific, # R70007) were cultured in DMEM (Nacalai Tesque, #08459-64) supplemented with 10% FBS (Biosera, #FB-1285/500), MEM Non-Essential Amino Acids Solution (Thermo Fisher Scientific, #11140-050), 1 mM sodium pyruvate (Sigma-

Aldrich, #S8636), and 1 mM L-glutamine (Thermo Fisher Scientific, #25030-081) in a humidified incubator at 37 °C with 5% $CO_2$. The cells at <30 passages were used for all the experiments. The cells at different passages were used for preparing each biological replicate. In the fluorescein induction experiment, the cells were treated with fluorescein diacetate (Tokyo Chemical Industry, #F0240) dissolved in ethanol at a final concentration of 50 or 100 µg/mL, 16 h after transfection. The cells were cultured in the media with fluorescein until the subsequent analysis and were washed with PBS before the analysis. In the heat shock experiment, the cells were exposed to 42 °C for 1 h, 16 h after transfection.

### Plasmid transfection
293FT cells with ~80–90% confluency were used for the plasmid transfection. 293FT cells were seeded in 24-well plates at $1 \times 10^5$ cells per well the day before transfection. The cells were transfected with a mixture of the plasmids using 2.0 µL of Lipofectamine 2000 (Thermo Fisher Scientific, #11668019) following the manufacturer's instructions. The amounts of transfected plasmids are listed in Supplementary Table 4.

### Generation of stable cell lines using CRISPR-Cas9
The stable cell lines expressing EGFP-GCN4 or EGFP were generated using CRISPR-Cas9[63]. The genes for each protein were cloned into a donor plasmid with homology arms targeting the endogenous *AAVS1* locus. 293FT cells were seeded in 24-well plates at $1 \times 10^5$ cells per well the day before transfection. The cells were transfected with 1 µg of SpCas9 plasmid (Addgene plasmid #41815), 1 µg of *AAVS1*-targeting sgRNA plasmid (Addgene plasmid #41818), and 1 µg of the donor plasmid using 2.0 µL of Lipofectamine 2000 following the manufacturer's instructions. Two days after the transfection, the cells were trypsinized and seeded into six-well plates with the medium containing 1 µg/mL puromycin (Invivogen, #ant-pr-1). The fluorescence-positive cells were then sorted and enriched using FACSymphony S6 cell sorter (BD Biosciences). The sorted cells were maintained in the medium containing 0.5 µg/mL puromycin.

### Luciferase assay
All the luciferase assays were conducted using Dual-Glo Luciferase Assay System (Promega, #E2920) following the manufacturer's instructions. 293FT cells were harvested and lysed 2 days after transfection. The activities of firefly luciferase and Renilla luciferase were measured using GloMax Navigator Microplate Luminometer (Promega). Relative luciferase units (RLU) were calculated by normalizing firefly luciferase activity to Renilla luciferase activity in each sample. Each biological replicate contained two technical replicates.

### Fluorescent reporter assay
The cells were imaged 2 days after transfection using CellVoyager CQ1 (Yokogawa Electric Corporation). Each image was processed using ImageJ software (National Institutes of Health) and its plugin (https://github.com/yfujita-skgcat/image_converter). For quantification, the total integrated intensity of the tdTomato fluorescence-positive cells in each image was quantified as fluorescence intensity using ImageJ. Each biological replicate contained two technical replicates. To calculate the mean tdTomato fluorescence intensity, the cells were analyzed using CytoFLEX S Flow Cytometer (Beckman Coulter), and the acquired data were analyzed using FlowJo software (BD Biosciences).

### CRISPR-mediated gene knockout
In the CRISPR knockout experiments, the stable cell lines; EGFP-GCN4 cells and EGFP cells, were used. The cells were seeded in 24-well plates and transfected with corresponding plasmids listed in Supplementary Table 4. Three days after the transfection, the cells were trypsinized

and seeded into six-well plates with the medium containing 0.5 µg/mL puromycin. Seven days after the transfection, the cells were analyzed using CytoFLEX S Flow Cytometer. The acquired data were analyzed to calculate the EGFP-negative population using FlowJo software and the "flowCore" package of R (https://bioconductor.org/packages/release/bioc/html/flowCore.html).

### In silico protein structure prediction
The structure of the CDR-grafted variable region of the anti-FLAG antibody was predicted using ColabFold[64], a Google Colab-based protein structure prediction with AlphaFold2 and MMseqs2, with default settings. The predicted structures were visualized and aligned to the original variable region of the anti-FLAG antibody (PDB ID: 7BG1) using PyMOL software.

### Reporting summary
Further information on research design is available in the Nature Portfolio Reporting Summary linked to this article.

## Data availability
All the data supporting this study are available within the main text and the Supplementary Information file. Cited crystal structure data are available in the Protein Database (1HLL, 1P4B, 1QLN, 2Y0G, 3ADF, 4PHY, 5A2Q, 5VIV, 7BG1) and the AlphaFold Protein Structure Database (AF-P0DMV8-F1). Materials such as plasmids are available from the corresponding author upon request. Source data are provided with this paper.

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

## Acknowledgements

We thank Dr. Yoshihiko Fujita (Kyoto University), Dr. Tatsuyuki Yoshii (Kyoto University), Dr. Shin Kaneko (Kyoto University), Dr. Atsutaka Minagawa (Kyoto University), and Dr. Yoshihiro Shimizu (RIKEN) for helpful discussions, and Dr. Shunsuke Kawasaki (Kyoto University) and Dr. Moe Hirosawa (Kyoto University) for supplying materials. Dr. Kanae Mitsunaga (Kyoto University) for technical assistance, Dr. Kelvin K. Hui (Kyoto University) and Maya Lopez (Imperial College London) for proofreading the manuscript, and Yuko Kono, Hiromi Takemoto, and Yoshiko Ogawa for their administrative support, and Dr. Dan Liu (National Institute of Technology, Ariake College) for encouragement throughout the research process. S.K. was supported by Scholarships from Iwadare Scholarship Foundation and Honjo International Scholarship Foundation. This work was supported by JST SPRING, Grant Number JPMJSP2110. This work was also supported by the JSPS KAKENHI (Grant Numbers JP20H05626 and 20H05701) and iPS Cell Research Fund from Center for iPS Cell Research and Application, Kyoto University.

## Author contributions

S.K. performed all the experiments. S.K., H.O., and H.S. designed experimental strategies and wrote the paper.

## Competing interests

The authors declare no competing interests.
