## [Peer Review File · Nature Communications]

Reviewers' Comments:

Reviewer #1:

Remarks to the Author:

This paper presents an interesting study on the development of a target-dependent RNA polymerase (TdRNAP), by fusing split phage T7 RNA polymerase with the VH and VL domains of the antibodies. The authors showed that by fusing the domains of the antibody can induce the assembly of split RNAP into a functional RNAP in the presence of their respective molecular targets.

The split T7 RNAP was based on an earlier paper, ref (15), which created the proximity dependent split T7 RNAP sensor. The authors overcome the earlier limitation in which regulating the activity of earlier reported split T7 RNAP sensors were limited as there was a lack of inducible dimerization proteins. The authors address this by exploring the use of identified antibodies. While it wasn't mentioned by the authors in the paper, there were also reports that fused split T7 RNAP with proteins for regulation of the activity of split T7 RNAP by light and heat for prokaryotes. The approach presented in this paper is different from these earlier works.

The authors demonstrated well that the TdRNAPs were able to respond to different molecule targets including peptides, proteins, RNA and small molecules depending on the antibodies fused to the split T7 RNAP and induce transcription in human cells. The authors have used several earlier identified antibodies to demonstrate their approach. This showed that the approach is quite versatile. The authors also tested the specificity of the TdRNAPs fused to different antibodies and showed that the fused antibody variable regions can specifically recognize their respective targets well. The paper demonstrated the utility of the TdRNAPs for cell specific genome editing dependent on the target gene product. Overall, the paper was well written. I find that the approach interesting and has expanded our ability to control gene expression in mammalian cells. I have a few comments/questions.

1. It seems that a limitation lies in finding orthogonal T7 promoters/RNAP if multiplexed gene expression is required, e.g. expression of different genes upon detection of different molecules. Could the author comment on this?

2. Line 61 to 63. References should be provided to support the statements, particularly regarding earlier studies that have studied splitting of T7 RNA polymerases.

Reviewer #2:

Remarks to the Author:

The manuscript describes construction of a chemically induced dimerization (CID) system that controls activity of split T7 RNA polymerase. The CID system for different analytes are constructed using fragments of antibody targeting the desired analyte. The system exploits the ability of the ligand to induced dimerization of heavy and light chain fragments. While this phenomenon was precisely observed in vitro and was used to construct protein biosensors it was not clear how useful this approach was due to inferior biophysical properties.

The presented data demonstrates that such systems work surprisingly well in the mammalian cells and could be adopted to different analytes. In retrospect this is easy to rationalise as mammalian cytosol has high chaperon activity and even marginally stable system are able to operate there. For instance, uniRap fusion of FRB and FKBP that is regulated by rapamycin works well in mammalian cells but there are no reports on its in vitro analysis or its application to bacterial systems.

Discussion of this point would be important for framing the results.

I think the authors make a strong case for general utility of their systems and while I suspect that there will be easier to apply it to proteins than to other classes of ligands presented examples are sufficiently convincing. The ability of build complex genetic circuits is using this approach is very impressive.

The paper would be even stronger if the authors benchmarked a couple of their switches against another well characterised CID system mentioned in the introduction. This would help to put what appears a modest dynamic range of the system into context.

The paper is well written and while it may contains minor technical flaws related to mammalian cells experiments that are outside of my expertise I think given the clarifications able it should merit publications.

We thank all reviewers for their positive feedback and valuable comments to improve our manuscript. We have addressed the referee's comments with point-by-point responses and revised the manuscript. We believe that all suggestions from the reviewers have been adequately addressed and that these changes have significantly improved the novelty, readability, and clarity of our manuscript.

Point-By-Point Response:

Reviewer #1

1. This paper presents an interesting study on the development of a target-dependent RNA polymerase (TdRNAP), by fusing split phage T7 RNA polymerase with the VH and VL domains of the antibodies. The authors showed that by fusing the domains of the antibody can induce the assembly of split RNAP into a functional RNAP in the presence of their respective molecular targets.

The split T7 RNAP was based on an earlier paper, ref (15), which created the proximity dependent split T7 RNAP sensor. The authors overcome the earlier limitation in which regulating the activity of earlier reported split T7 RNAP sensors were limited as there was a lack of inducible dimerization proteins. The authors address this by exploring the use of identified antibodies. While it wasn't mentioned by the authors in the paper, there were also reports that fused split T7 RNAP with proteins for regulation of the activity of split T7 RNAP by light and heat for prokaryotes. The approach presented in this paper is different from these earlier works.

Response:

We thank the reviewer for bringing up this important point. As the reviewer noted, there were other approaches to regulate the split T7 RNAP activity. To clearly distinguish the difference between our work and earlier works, we have added a sentence about the light-inducible split RNAP system in the introduction. Additionally, we have changed "PPI inducers" to "PPI inducible molecules" to clarify that our study focuses on molecular-responsive gene regulation.

-Lines 65-72:

This evolved split RNAP has enabled inducible RNA transcription that is regulated by the protein-protein interaction (PPI) between fused proteins. The light and chemical-inducible dimerization domains allowed the split RNAP to be activated by blue light and small molecules, respectively^{18,19}. Thus, this split RNAP holds great potential to control gene expression in response to intracellular molecules without relying on cellular machinery. However, the available PPI inducible molecules for regulating split RNAP activity have been limited to commonly used small molecules, such as rapamycin and abscisic acid, due to a lack of inducible dimerization proteins^{18,19}.

2. The authors demonstrated well that the TdRNAPs were able to respond to different molecule targets including peptides, proteins, RNA and small molecules depending on the antibodies fused to the split T7 RNAP and induce transcription in human cells. The authors have used several earlier identified antibodies to demonstrate their approach. This showed that the approach is quite versatile. The authors also tested the specificity of the TdRNAPs fused to different antibodies and showed that the fused antibody variable regions can specifically recognize their respective targets well. The paper demonstrated the utility of the TdRNAPs for cell specific genome editing dependent on the target gene product. Overall, the paper was well written. I find that the approach interesting and has expanded our ability to control gene expression in mammalian cells. I have a few comments/questions.

1. It seems that a limitation lies in finding orthogonal T7 promoters/RNAP if multiplexed gene expression is required, e.g. expression of different genes upon detection of different molecules. Could the author comment on this?

Response:

This is an important point for the design of multiplex biosensors and multilayer gene circuits. Previously, the five orthogonal T7 RNAP variant/promoter pairs were developed by directed evolution (ref 48 in the revised manuscript). We used one of them, CGG RNAP/CGG promoter pair, to show orthogonal and multiplex gene regulation using GCN4-dependent CGG RNAP and EGFP-dependent T7 RNAP (Fig. 4d). In addition, the paper from the Dickinson lab (Pu et al., *J. Am. Chem. Soc.* 2017) has already used the other orthogonal T7

RNAP/promoter pairs in the same split format that we used and demonstrated orthogonal and multiplex gene regulation depending on the interaction between fusion proteins. According to these facts, we believe that at least these five orthogonal T7 RNAP/promoter pairs, in addition to the wild-type T7 RNAP/promoter pair, can be used to design orthogonal TdRNAPs and control six different genes upon the detection of six different molecules recognized by their corresponding antibodies. Similarly, other bacteriophage single-subunit RNAPs, such as SP6 RNAP, could also be evolved to develop proximity-dependent RNAPs and applied as new orthogonal TdRNAPs.

We have cited the paper from the Dickinson lab (Pu et al., *J. Am. Chem. Soc.* 2017) and a review paper about bacteriophage RNAPs (Shutt, T. E. & Gray, M. W., *Trends Genet.* 2006) and further discussed the orthogonal and multiplexed gene regulation in the discussion as follows:

-Lines 376-383: (The addition of the detailed sentence and ref. 49 and 50)

Second, multiple and orthogonal TdRNAPs can be easily designed using reported T7 RNAP variants. The five orthogonal T7 RNAP variants and their respective promoters have been developed by directed evolution and are also available for split RNAP architecture^{48,49}. These orthogonal RNAP and promoter pairs can be combined with different antibodies to design orthogonal TdRNAPs. Additionally, other single-subunit RNAPs derived from bacteriophages, such as SP6 RNAP, could also be used to design new orthogonal TdRNAPs by molecular evolution⁵⁰. TdRNAPs designed with these evolved and orthogonal RNAPs could allow the multiplex molecular response and orthogonal gene regulation.

3. 2. Line 61 to 63. References should be provided to support the statements, particularly regarding earlier studies that have studied splitting of T7 RNA polymerases.

Response:

We thank the reviewer for the suggestion. We have added the following references to the indicated lines of the manuscript.

-Lines 61-62: (the addition of ref. 15)

Bacteriophage T7 RNA polymerase (T7 RNAP) exhibits high transcriptional activity and is capable of synthesizing RNA from DNA templates without the need for additional factors [Wang et al., *Biotechnol. Adv.* 2018].

-Lines 62-64: (the addition of ref. 16 and 17)

The T7 RNAP can be divided into N- and C-terminal fragments that spontaneously assemble into functional RNAP [Ikeda et al., *J. Biol. Chem.* 1987], [Shis et al., *Proc. Natl. Acad. Sci. U S A* 2013].

Reviewer #2

1. The manuscript describes construction of a chemically induced dimerization (CID) system that controls activity of split T7 RNA polymerase. The CID system for different analytes are constructed using fragments of antibody targeting the desired analyte. The system exploits the ability of the ligand to induced dimerization of heavy and light chain fragments. While this phenomenon was precisely observed in vitro and was used to construct protein biosensors it was not clear how useful this approach was due to inferior biophysical properties.

The presented data demonstrates that such systems work surprisingly well in the mammalian cells and could be adopted to different analytes. In retrospect this is easy to rationalise as mammalian cytosol has high chaperon activity and even marginally stable system are able to operate there. For instance, uniRap fusion of FRB and FKBP that is regulated by rapamycin works well in mammalian cells but there are no reports on its in vitro analysis or its application to bacterial systems.

Discussion of this point would be important for framing the results.

Response:

We thank the reviewer for the interesting suggestion. As the reviewer pointed out, the high chaperone activity in mammalian cells may contribute to efficient and stable protein folding and the TdRNAP activity. We agree

with this point, as our result also indicated the importance of the structural stability of the VH and VL domains for the TdRNAP activity.

We have discussed the possible effect of mammalian chaperone activity on our system by adding the following sentence at the indicated location in the result section as follows:

-Lines 204-208: (The addition of the following sentence)

These findings show that each VH and VL domain retains its target specificity in mammalian cells, indicating that the CDR loop structures of the antibody variable regions are properly folded and stabilized. This suggests that the stable framework region and the molecular chaperone of mammalian cells may contribute to the stabilized CDR loop structures and target specificity.

2. I think the authors make a strong case for general utility of their systems and while I suspect that there will be easier to apply it to proteins than to other classes of ligands presented examples are sufficiently convincing. The ability of build complex genetic circuits is using this approach is very impressive. The paper would be even stronger if the authors benchmarked a couple of their switches against another well characterised CID system mentioned in the introduction. This would help to put what appears a modest dynamic range of the system into context.

Response:

We thank the reviewer for the suggestion. We agree with the reviewer that our system exhibited a modest dynamic range compared with existing CID systems, such as the rapamycin-responsive biosensor. Unfortunately, direct comparisons between existing CID systems and our system are challenging because there are no antibodies with publicly available amino acid sequences targeting the same molecules as the well-characterized CID systems, such as rapamycin. Similarly, no existing CID systems can respond to the same molecules (i.e., protein/peptide or RNA) used in this study (a testament to the novelty of this work). Nevertheless, we think it is vital to discuss the limitations of our system, as highlighted by the reviewer, and provide a clear assessment of the pros and cons when comparing our system to conventional CID systems.

We have further discussed the limitations and the differences between our system and conventional systems in the discussion.

Lines 371-375:

First, it is possible to use VH and VL domains as inducible dimerization domains without selecting the split sites of ligand-binding proteins and screening dimerization domains. This design principle allows for the easy targeting of various intracellular molecules compared with conventional platforms, enabling gene regulation based on cellular conditions. This advantage greatly expands the application range for molecular-inducible gene regulation systems.

Lines 389-392:

To achieve a high dynamic range, similar to well-characterized inducible dimerization systems such as rapamycin-inducible systems^{18,19,53}, further improvement and optimization of the antibody-based dimerization system are likely necessary.

3. The paper is well written and while it may contains minor technical flaws related to mammalian cells experiments that are outside of my expertise I think given the clarifications able it should merit publications.

Response:

We thank the reviewer for the important suggestion. We have added more detailed descriptions in the following locations within the methods section. We also corrected the grammatical and typing errors and changed the sentences as follows:

-Lines 449-451: (sentence added)

The cells were cultured in the media with fluorescein until the subsequent analysis and were washed with PBS before the analysis.

-Line 455: (sentence added)

293FT cells with approximately 80-90% confluency were used for the plasmid transfection.

-Line 483: (software source added)

ImageJ software (National Institutes of Health)

-Lines 497-498: (software source added)

the “flowCore” package of R (<https://bioconductor.org/packages/release/bioc/html/flowCore.html>).

-Line 130: (long forms of FKBP and FRB added)

Before: FKBP and FRB

After: FK506 binding protein (FKBP) and FKBP-rapamycin binding domain (FRB)

-Line 22: (grammatical error corrected)

Before: provides

After: provide

-Line 165: (typing error corrected)

Before: intracellular

After: intramolecular

-Line 417: (grammatical error corrected)

Before: respond

After: responds

-Line 428: (typing error corrected)

Before: kunkel

After: Kunkel

-Line 430: (grammatical error corrected)

Before: The gene for evolved T7 RNAP N-terminal fragment

After: The gene for the evolved T7 RNAP N-terminal fragment

-Line 431: (typing error corrected)

Before: mutageneses

After: mutagenesis

-Lines 247-249: (We changed the description of the results shown in Fig. 3d and Fig. 3e. The previous sentence mentioned only the result of the fluorescent reporter assay, not the luciferase assay. We have changed both “tdTomato fluorescence” and “tdTomato expression” to “reporter expression” in the indicated sentence.)

Before: Treatment with fluorescein led to a dose-dependent increase in tdTomato fluorescence, while the treatment with the vehicle only mildly induced tdTomato expression (Fig. 3d, e).

After: Treatment with fluorescein led to a dose-dependent increase in reporter expression, while the treatment with the vehicle only mildly induced reporter expression (Fig. 3d, e).

-Supplementary Table 2 (typing error corrected)

Before: CGG RANPc (C-terminal CGG RNAP fragment)

After: CGG RNAPc (C-terminal CGG RNAP fragment)

-Supplementary Table 4, Fig. 2c and 2d (typing error corrected)

Before1: Amount of pUC19 (ng): up to 200

Before2: Total amount (ng): 200

After1: Amount of pUC19 (ng): up to 300

After2: Total amount (ng): 300

-Acknowledgements (people added)

Dr. Shin Kaneko and Dr. Atsutaka Minagawa

Reviewers' Comments:

Reviewer #1:

Remarks to the Author:

I find the comments well addressed. Thanks

Reviewer #2:

Remarks to the Author:

The authors adequately responded to my minor criticism and I believe that the paper should be published in its current form. I congratulate the authors on a very interesting and potentially impactful study.